# *CYP2D6* and *CYP2C19* Variant Coverage of Commercial Antidepressant Pharmacogenomic Testing Panels Available in Victoria, Australia

**DOI:** 10.3390/genes14101945

**Published:** 2023-10-16

**Authors:** Malcolm Forbes, Mal Hopwood, Chad A. Bousman

**Affiliations:** 1The Institute for Mental and Physical Health and Clinical Translation (IMPACT), School of Medicine, Barwon Health, Deakin University, Geelong, VIC 3220, Australia; 2Department of Psychiatry, University of Melbourne, Parkville, VIC 3050, Australia; mhopwood@unimelb.edu.au (M.H.); chad.bousman@ucalgary.ca (C.A.B.); 3Department of Medical Genetics, University of Calgary, Calgary, AB T2N 4N2, Canada

**Keywords:** pharmacogenetics, psychiatry, personalized medicine, *CYP2D6*, *CYP2C19*

## Abstract

Pharmacogenomic (PGx) testing to inform antidepressant medication selection and dosing is gaining attention from healthcare professionals, patients, and payors in Australia. However, there is often uncertainty regarding which test is most suitable for a particular patient. Here, we identified and evaluated the coverage of *CYP2D6* and *CYP2C19* variants in commercial antidepressant PGx testing panels in Victoria, a large and ethnically diverse state of Australia. Test characteristics and star alleles tested for both genes were obtained directly from pathology laboratories offering PGx testing and compared against the Association of Molecular Pathology’s recommended minimum (Tier 1) and extended (Tier 2) allele sets. Although all tests covered the minimum recommended alleles for *CYP2C19*, this was not the case for *CYP2D6*. This study emphasizes that PGx tests might not be suitable for all individuals in Australia due to the limited range of star alleles assessed. Inadequate haplotype coverage may risk misclassification of an individual’s predicted metabolizer phenotype, which has ramifications for depression medication selection and dosage. This study underscores the urgent need for greater standardization in PGx testing and emphasizes the importance of considering genetic ancestry when choosing a PGx testing panel to ensure optimal clinical applicability.

## 1. Introduction

A significant portion of the variability in drug response can be explained by genetic variation. Pharmacogenomic (PGx) testing is a tool for detecting this genetic variation, which can then inform medication selection and dosing. To date, PGx testing has predominately focused on genes encoding drug-metabolizing enzymes within the cytochrome P450 (CYP) family [1]. This is particularly the case for antidepressant medications for which two genes (*CYP2D6* and *CYP2C19*) are the foundation of several dosing guidelines developed by the Clinical Pharmacogenetics Implementation Consortium (CPIC) [2,3,4].

The development of these guidelines has, in part, stimulated patient and physician demand for PGx testing, and as a result, laboratories have begun to add PGx testing to their list of tests offered. In fact, the number of laboratories offering PGx testing is growing, and as such, healthcare providers tasked with ordering these tests often have numerous laboratories to choose from and are typically uncertain which testing laboratory’s panel is best suited for their patients. Although there are several factors to consider when selecting a PGx testing laboratory, arguably one of the most important considerations is the laboratory’s testing panel coverage. Current regulatory standards for laboratories do not dictate what genes or variants within genes should or should not be tested. As a result, variability in PGx testing panel coverage across laboratories is common [5].

Both *CYP2D6* and *CYP2C19* are highly polymorphic, with over 170 and 35 star alleles catalogued by the Pharmacogene Variation (PharmVar) Consortium, respectively [6]. A star (*) allele, or haplotype, refers to a combination of single nucleotide variants (SNVs) and structural variants (SVs) inherited together. The combination of two haplotypes (one from each parent) forms the diplotype, which is used to assign a metabolizer phenotype (ultrarapid, rapid, normal, intermediate, or poor). In general, the probability of misassigning a metabolizer phenotype decreases as the number of star alleles tested by a laboratory increases. The reasoning behind this is that the failure to detect a star allele will result in the assignment of the reference allele (*1), which is presumed to have normal function. Thus, laboratories testing fewer star alleles are more likely to assign the *1 allele and are at greater risk for false negatives (i.e., assigning normal metabolizer status to a patient that is an ultrarapid, rapid, intermediate, or poor metabolizer).

The consequences of false negatives are non-trivial, as antidepressant plasma concentrations at standard doses can vary substantially between *CYP2D6* and *CYP2C19* metabolizer phenotypes [7]. Thus, a poor metabolizer that is inaccurately classified as a normal metabolizer may reach supratherapeutic levels of an antidepressant at standard dosing and be at greater risk of side effects, whereas an ultrarapid metabolizer classified as a normal metabolizer may not reach therapeutic levels if treated with standard dosing. As such, accurate assignment of metabolizer phenotypes is imperative for the clinical efficacy of PGx testing. This study sought to evaluate *CYP2D6* and *CYP2C19* allele coverage of commercial pharmacogenomic test panels available in Victoria, Australia, and compare this coverage to allele selection recommendations for *CYP2D6* and *CYP2C19* developed by the Association of Molecular Pathology [8,9].

## 2. Materials and Methods

In September 2023, one author (MF) contacted all pathology laboratories offering PGx testing in Victoria, Australia. Each laboratory was asked to provide details on their mental health or antidepressant PGx testing panel, including the specific genes and corresponding variants/star alleles tested as well as their testing methodology, turnaround time (i.e., time from receipt of sample to test result), and cost of the test (as patients bear the full cost of testing). Although every laboratory included several genes on their panels, only *CYP2D6* and *CYP2C19* were evaluated in-depth for the current study as these two genes are the most relevant to antidepressant prescribing.

*CYP2D6* and *CYP2C19* star alleles for each panel were assessed against allele selection recommendations developed by the Association of Molecular Pathology (AMP) [8,9]. These recommendations are organized into two tiers. Tier 1 represents the minimum recommended set of alleles that should be included on a testing panel. All Tier 1 alleles meet three criteria: (1) have a well-characterized effect on the function of the protein and/or gene expression, (2) have an appreciable minor allele frequency in a population/ethnicity group, and (3) have publicly available reference materials that provide testing laboratories the ability to assess the analytical validity of their assays. Tier 2 represents an extended panel of alleles that meet at least one but not all three criteria. In our evaluation, we assessed *CYP2D6* and *CYP2C19* allele coverage against AMP’s Tier 1 and Tier 2 recommendations. We also cross-referenced and provided the frequency of each allele in eight biogeographical groups: Sub-Saharan African, African American/Afro-Caribbean, European, Near Eastern, East Asian, South/Central Asian, American, and Oceanian as defined by the Pharmacogenomics Knowledge Base (PharmGKB) [10].

## 3. Results

### 3.1. Test Characteristics

Four laboratories offering PGx testing were identified in Victoria, Australia. Two laboratories offered specific ‘mental health’ PGx tests, and two offered general PGx screening tests. As shown in Table 1, three of the four laboratories used the Agena Biosciences MassARRAY System for performing genotyping. The number of genes tested ranged from five to eleven genes, although most panels included one or more genes without PGx-based prescribing guidelines developed by the Clinical Pharmacogenetics Implementation Consortium [11] or the Dutch Pharmacogenetics Working Group [12]. All four laboratories included *CYP2C19* and *CYP2D6*. Testing turnaround time ranged from 5 to 10 business days, and test costs ranged from AUD $149 to $197.

### 3.2. CYP2C19 and CYP2D6 Allele Coverage

Table 2 summarizes *CYP2C19* and *CYP2D6* allele coverage for the four laboratories. For *CYP2C19*, all tests included the recommended minimum (Tier 1) allele set (*2, *3, *17). Three labs included at least one Tier 2 allele on their panel. MyDNA tested the CYP2D6*9 variant with a frequency of 2.7% in Sub-Saharan African populations. Two of the labs (Sonic Genetics and Incite Genomics) included five Tier 2 alleles (*4, *5, *6, *7, *8), with frequencies ranging from 0% to 0.3% across the eight PharmGKB biogeographical groups.

For *CYP2D6*, three laboratories (MyDNA, Sonic Genetics, and Incite Genomics) included the recommended minimum (Tier 1) allele set (*2, *3, *4, *5, *6, *9, *10, *17, *29, *41, *xN). Tier 2 allele coverage varied by lab and ranged between two and five alleles. All the labs included one to four *CYP2D6* alleles on their panels that are not included in the AMP Tier 1 or Tier 2 allele sets, with allele frequencies ranging from 0% to 2.7% across the eight PharmGKB biogeographical groups.

## 4. Discussion

Pharmacogenomic testing (PGx) is increasingly relevant in the pharmacological management of depression, a disorder with a high degree of variability in treatment response [13,14]. There is evidence to support the cost-effectiveness of PGx testing in depression [15], with the demonstrated potential to shorten the time to reach clinical response and remission [16,17,18] and reduce adverse effects [19,20]. However, inadequate standardization of tests and insufficient coverage of alleles present in diverse populations limit PGx utility.

Our results support the notion that PGx testing panels differ from lab to lab, with variable coverage of *CYP2C19* and *CYP2D6* alleles. In fact, we found that no two commercially available tests in Victoria, Australia, assessed for the same set of variants. The lack of standardization across labs could result in clinically meaningful differences. For example, the *CYP2D6**29 decreased function allele was not tested in all panels. This allele is uncommon in individuals of European ancestry (0.1%) but is common among those of African ancestry (8.7–10.8%). The omission of this allele means that an individual with this variant could be assigned a metabolizer status discordant with their “true” status (e.g., assigned a normal metabolizer when, in fact, they are an intermediate metabolizer) [21], highlighting the importance of considering ancestry when ordering PGx testing [22,23].

There are significant differences in *CYP2D6* and *CYP2C19* variants between biogeographical populations; however, there has been limited investigation of allelic variation in non-European populations [24]. Most PGx knowledge is derived from European and East Asian male populations, with the risk that “understudied populations with more diverse haplotype frequencies are therefore more likely to be affected by imprecision when applying pharmacogenomic annotations to dosage administration” [25]. There is evidence outside of psychiatry that demonstrates this point. For instance, most warfarin dosing algorithms are based on polymorphisms that are prevalent in European populations but relatively rare in African populations [26].

Victoria is Australia’s second most populous state and has an ethnically diverse population. Over 15% of Victoria’s population are of Chinese, Indian, Vietnamese, Sri Lankan, Filipino, Māori, Samoan, or Korean ancestry, and 6.5% are Aboriginal and/or Torres Strait Islanders, the Indigenous people of Australia. Immigration to Australia from Africa has increased over the past decade [27], which may increase allelic variation, given that African populations have the highest *CYP2D6* haplotype diversity [28]. Despite an ethnically diverse population, there is limited published information about allelic diversity at *CYP2C19* and *CYP2D6* in Australia. One study that used a convenience sample of 5408 patients found that 96% of the sample had at least one actionable variant. However, this study included mostly participants of a European background. There is also limited information about allelic diversity of *CYP2C19* and *CYD2D6* within Indigenous populations outside of northern Australia [29,30,31] to inform testing in this group. Further complicating matters, while it is important to consider self-reported race/ethnicity in PGx testing, there are considerable limitations in self-reported race/ethnicity as these often involve some element of social construction and do not reliably capture an individual’s genetic ancestry [32]. This is particularly the case for individuals with higher levels of genetic admixture [33].

There is an urgent need for greater standardization of PGx testing. While there are only four commercial laboratories providing testing in Victoria, Australia, there are over 75 laboratories in the USA [21] and 13 in Canada [5] that offer pharmacogenomic testing, with considerable variation in *CYP2D6* and *CYP2C19* alleles tested across these laboratories [34]. Our study suggests a need for greater transparency about the limitations of current PGx tests in ethnically diverse populations, including the possibility of phenotype misclassification. Furthermore, it should be noted that interindividual differences in drug response are a mix of genetic as well as environmental and pathophysiological factors, including age, sex, hormonal status, dietary intake, alcohol, and other illicit and prescription drug use [35,36,37]. Improvements in technological capability, including next-generation sequencing and computational prediction, may allow for the discovery of rare or novel variants and the integration of non-genetic factors that can enhance individualized drug prescription [38].

## 5. Conclusions

There is considerable variation in PGx testing in Australia, which has ramifications for the external validity of these tests [39]. Individuals considering such tests should consider the allelic variants tested and the prevalence of specific variants in biogeographical populations. With future PGx tool development and broader representation of all biogeographical populations, we hope the potential benefits of this technology may be realized to improve treatment for all individuals with serious mental illness.

## Figures and Tables

**Table 1 genes-14-01945-t001:** Characteristics of commercial pharmacogenetic tests to guide antidepressant dosing available in Victoria, Australia.

Characteristic	Australian Clinical Labs	MyDNA/Genomic Diagnostics	Sonic Genetics	Incite Genomics
**Name**	Comprehensive Gene Panel	Mental Health Medication Test	Pharmacogenomic Screen	Amplis Evo Mental Health
**Genotyping platform**	Agena MassARRAY	Thermofisher and Taqman Real-Time Open Array	Agena MassARRAY	Agena MassARRAY
**Genes tested**	***CYP2C19***, ***CYP2D6***, ***CYP2C9***, ***CYP3A4***, ***CYP3A5***, *CYP1A2*, ***SLCO1B1***, ***VKORC1***	***CYP2C19***, ***CYP2D6***, ***CYP2C9***, *CYP1A2*, ***CYP3A4***	***CYP2C19***, ***CYP2D6***, ***CYP2C9***, *CYP1A2*, ***CYP3A4***, ***CYP3A5***, *ABCB1*, *OPRM1*, ***SLCO1B1***, ***VKORC1***	***CYP2C19***, ***CYP2D6***, ***CYP2C9***, ***CYB2B6***, *CYP1A2*, ***CYP3A4***, ***CYP3A5***, *ABCB1*, *ABCC1*, *ABCG2*, ***UGT1A1***
**Turnaround time (maximum)**	10 business days	10 business days	10 business days	5 business days
**Cost (AUD)**	190	149	197	195

Bolded genes are those with pharmacogenomic-based prescribing guidelines developed by the Clinical Pharmacogenetics Implementation Consortium or the Dutch Pharmacogenetics Working Group.

**Table 2 genes-14-01945-t002:** *CYP2C19* and *CYP2D6* Tier 1 and 2 allele coverage by Victoria labs and frequencies in PharmGKB biogeographical groups.

	Commercial PGx Testing Labs in Victoria, Australia				PharmGKB Biogeographical Groups *							
	Australian Clinical Labs	MyDNA/Genomic Diagnostics	Sonic Genetics/Melbourne Pathology	Incite Genomics	Sub-Saharan African	African American/Afro-Caribbean	European	Near Eastern	East Asian	South/Central Asian	Americas	Oceanian
** *CYP2C19* **												
**Tier 1**												
*2	X	X	X	X	15.7%	18.1%	14.7%	12.0%	28.4%	27.0%	12.1%	61.0%
*3	X	X	X	X	0.3%	0.3%	0.2%	1.6%	7.2%	1.6%	0.0%	14.6%
*17	X	X	X	X	17.3%	20.7%	21.5%	19.1%	2.1%	17.1%	8.6%	5.7%
**Tier 2**												
*4			X	X	0.0%	0.0%	0.2%	0.0%	0.0%	0.0%	0.0%	–
*5			X	X	0.0%	0.0%	0.0%	0.0%	0.3%	0.0%	0.0%	–
*6			X	X	0.0%	0.0%	0.0%	0.0%	0.1%	0.0%	–	–
*7			X	X	0.0%	0.0%	0.0%	–	0.0%	0.0%	–	–
*8			X	X	0.0%	0.1%	0.3%	0.0%	0.0%	0.0%	0.0%	–
*9		X			2.7%	1.4%	0.1%	–	0.0%	–	–	–
*10					0.0%	0.3%	0.0%	0.0%	0.0%	–	–	–
*35					3.2%	1.6%	0.0%	–	0.0%	–	–	–
** *CYP2D6* **												
**Tier 1**												
*2	X	X	X	X	17.4%	15.5%	18.5%	19.0%	11.9%	27.4%	21.7%	6.1%
*3	X	X	X	X	0.1%	0.3%	1.6%	0.4%	0.0%	0.1%	0.1%	0.1%
*4	X	X	X	X	2.9%	4.8%	18.5%	11.4%	0.5%	9.0%	10.2%	1.8%
*5	X	X	X	X	6.2%	5.4%	2.9%	1.8%	4.8%	4.2%	1.6%	3.5%
*6	X	X	X	X	0.0%	0.3%	1.1%	0.5%	0.0%	0.0%	0.3%	0.0%
*9	X	X	X	X	0.0%	0.4%	2.8%	0.4%	0.2%	0.2%	0.7%	0.0%
*10	X	X	X	X	4.9%	3.8%	1.6%	6.8%	42.8%	7.6%	1.5%	5.7%
*17	X	X	X	X	19.4%	16.9%	0.4%	3.1%	0.0%	0.0%	0.5%	0.1%
*29		X	X	X	10.8%	8.7%	0.1%	0.8%	0.0%	0.2%	0.2%	0.0%
*41	X	X	X	X	4.5%	3.7%	9.2%	15.4%	2.3%	11.9%	2.7%	3.2%
*xN	X	X	X	X	5.8%	6.0%	2.6%	7.5%	1.5%	1.4%	3.5%	11.9%
**Tier 2**												
*7		X	X	X	0.0%	0.0%	0.1%	0.3%	0.0%	0.7%	0.3%	0.0%
*8	X	X	X	X	0.0%	0.0%	0.0%	0.0%	0.0%	0.0%	0.1%	0.0%
*12		X	X	X	0.2%	0.1%	0.0%	0.0%	0.0%	0.0%	0.6%	–
*14	X	X	X	X	0.0%	0.0%	0.0%	–	0.5%	0.0%	0.0%	0.0%
*15			X	X	0.2%	0.0%	0.0%	0.0%	0.0%	0.0%	0.1%	0.0%
*21					0.0%	0.0%	0.0%	0.0%	0.4%	0.0%	0.0%	–
*31					0.0%	0.0%	0.1%	0.0%	0.0%	0.0%	0.6%	0.0%
*40					1.4%	0.5%	0.1%	0.0%	0.0%	0.0%	0.0%	0.0%
*42					0.1%	0.4%	0.0%	0.0%	0.0%	0.1%	0.0%	–
*49					0.0%	0.0%	0.0%	0.0%	1.0%	0.0%	0.0%	–
*56					0.2%	0.2%	0.1%	0.0%	0.0%	0.0%	0.0%	–
*59					0.0%	0.0%	0.4%	0.0%	0.0%	0.0%	0.1%	–
**Other**												
*11			X	X	0.0%	0.0%	0.0%	0.0%	0.0%	0.0%	0.0%	0.0%
*18			X	X	–	0.0%	0.0%	0.0%	0.1%	–	–	–
*19			X	X	0.0%	0.0%	0.0%	0.0%	0.0%	–	–	0.2%
*20				X	0.0%	0.0%	0.0%	0.0%	0.0%	0.0%	0.0%	0.0%
*36		X			0.4%	0.5%	0.0%	0.0%	1.1%	0.0%	0.0%	0.0%
*39	X				0.0%	2.0%	1.4%	2.7%	0.6%	0.3%	0.1%	0.6%
*114		X	X		–	–	–	–	0.1%	–	–	–

* Frequency data retrieved from the PharmGKB (https://www.pharmgkb.org/page/pgxGeneRef, accessed on 1 September 2023).

## Data Availability

All data available from this study have been included in the manuscript.

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
