# Peer review of "CYP2D6 and CYP2C19 Variant Coverage of Commercial Antidepressant Pharmacogenomic Testing Panels Available in Victoria, Australia"

_genes, 2023, doi:10.3390/genes14101945_

Round 1
Reviewer 1 Report
This is an excellently written manuscript addressing a major concern in mental health, namely the validity of pharmacogenetic testing to aid decision making in drug selection. Specifically this study examines the coverage of commercial tests available in Victoria, Australia, and their adherence to published guidelines. Background is fully covered, methodology is sound, results are clearly presented, and conclusions are supported. The only limitation that I see is the lack of a detailed comparison to other countries, which is only very briefly mentioned. If there are no available data for that, this should be stated.
Author Response
Response: We thank the Reviewer for their feedback. We have made note that there is a lack of detailed comparison between countries.
Revision (page 6, line 133-134): “However, inadequate standardization of tests and insufficient coverage of alleles present in diverse populations limit PGx utility.”
Reviewer 2 Report
The study is a comparison of variant coverage with respect to two genetically highly polymorphic drug metabolizing enzymes between four commercial genotyping providers in the Victoria state of Australia. While the overarching question is clinically relevant and generally of greater interest, this publication is strongly limited by the very narrow scope. It would have been of far greater interest and outreach had it covered a larger geographical region and/or a higher number of investigated polymorphic proteins. Nevertheless, it can inspire discussions and further research into this direction.
If such data is available, the introduction could be improved by providing information on the extent of genotyping and corresponding dosage adjustments in clinical care, i.e. how often are patients actually tested? Does the healthcare provider cover the costs or is it paid for by the patients?
Line 52: To my knowledge, there are only four metabolizer phenotypes: ultra-rapid, normal a.k.a extensive, intermediate, and poor. I have not heard of a fifth "rapid metabolizer" phenotype before.
The cited references should be more diverse and include more publications from groups other than the authors.
Altogether, the manuscript is very well written and easily comprehensible.
Author Response
The study is a comparison of variant coverage with respect to two genetically highly polymorphic drug metabolizing enzymes between four commercial genotyping providers in the Victoria state of Australia. While the overarching question is clinically relevant and generally of greater interest, this publication is strongly limited by the very narrow scope. It would have been of far greater interest and outreach had it covered a larger geographical region and/or a higher number of investigated polymorphic proteins. Nevertheless, it can inspire discussions and further research into this direction.
Response: We thank the Reviewer for their considered feedback. We accept that the focus of this article is narrow however the point that the article demonstrates, that there is considerable variation in PGx variant coverage, has broad implications.
If such data is available, the introduction could be improved by providing information on the extent of genotyping and corresponding dosage adjustments in clinical care, i.e. how often are patients actually tested? Does the healthcare provider cover the costs or is it paid for by the patients?
Response: Because PGx testing for antidepressant medication is not listed on the Medicare Benefits Schedule there are no readily available data to indicate the frequency of testing. There is no Government rebate within Australia and the cost of testing is borne by the patient. We have highlighted this in the manuscript.
Revision (page 2, line 72-76): “Each laboratory was asked to provide details on their mental health or antidepressant PGx testing panel including the specific genes and corresponding variants/star alleles tested as well as their testing methodology, turnaround time (i.e., time from receipt of sample to test result), and cost of the test (as patients bear the full cost of testing).”
Line 52: To my knowledge, there are only four metabolizer phenotypes: ultra-rapid, normal a.k.a extensive, intermediate, and poor. I have not heard of a fifth "rapid metabolizer" phenotype before.
Response: According to the Clinical Pharmacogenetics Implementation Consortium guidelines, CYP2D6 has four metabolizer phenotypes (ultrarapid, normal, intermediate, poor) and CYP2C19 has five phenotypes (ultrarapid, rapid, normal, intermediate, poor).
The cited references should be more diverse and include more publications from groups other than the authors.
Response: Further literature from a diverse range of research teams has been cited in the discussion of the manuscript.
Trivedi MH, Rush AJ, Wisniewski SR, Nierenberg AA, Warden D, Ritz L, et al. Evaluation of outcomes with citalopram for depression using measurement-based care in STAR*D: implications for clinical practice. Am J Psychiatry. 2006;163(1):28-40.
Cristancho P, Lenard E, Lenze EJ, Miller JP, Brown PJ, Roose SP, et al. Optimizing Outcomes of Treatment-Resistant Depression in Older Adults (OPTIMUM): Study Design and Treatment Characteristics of the First 396 Participants Randomized. Am J Geriatr Psychiatry. 2019;27(10):1138-52.
Morris SA, Alsaidi AT, Verbyla A, Cruz A, Macfarlane C, Bauer J, et al. Cost Effectiveness of Pharmacogenetic Testing for Drugs with Clinical Pharmacogenetics Implementation Consortium (CPIC) Guidelines: A Systematic Review. Clin Pharmacol Ther. 2022;112(6):1318-28.
Wang X, Wang C, Zhang Y, An Z. Effect of pharmacogenomics testing guiding on clinical outcomes in major depressive disorder: a systematic review and meta-analysis of RCT. BMC psychiatry. 2023;23(1):334.
Bunka M, Wong G, Kim D, Edwards L, Austin J, Doyle-Waters MM, et al. Evaluating treatment outcomes in pharmacogenomic-guided care for major depression: A rapid review and meta-analysis. Psychiatry Res. 2023;321:115102.
Vos CF, Ter Hark SE, Schellekens AFA, Spijker J, van der Meij A, Grotenhuis AJ, et al. Effectiveness of Genotype-Specific Tricyclic Antidepressant Dosing in Patients With Major Depressive Disorder: A Randomized Clinical Trial. JAMA Netw Open. 2023;6(5):e2312443.
Swen JJ, van der Wouden CH, Manson LE, Abdullah-Koolmees H, Blagec K, Blagus T, et al. A 12-gene pharmacogenetic panel to prevent adverse drug reactions: an open-label, multicentre, controlled, cluster-randomised crossover implementation study. Lancet. 2023;401(10374):347-56.
Altogether, the manuscript is very well written and easily comprehensible.
Response: Thank you.